# Machine learning for predicting cognitive decline within five years in Parkinson's disease: Comparing cognitive assessment scales with DAT SPECT and clinical biomarkers

**Arman Gorji**[1,2☉], **Ali Fathi Jouzdani**[ID][1,2☉]*

**1** Department of Neuroscience, School of Science and Advanced Technologies in Medicine, Neuroscience and Artificial Intelligence Research Group (NAIRG), Hamadan University of Medical Sciences, Hamadan, Iran, **2** USERN Office, Hamadan University of Medical Sciences, Hamadan, Iran

☉ These authors contributed equally to this work.
* Ali.fathi77@gmail.com, a.fathi@edu.umsha.ac.ir

## Abstract

### Objective

Parkinson's disease (PD) is an age-related neurodegenerative condition characterized mostly by motor symptoms. Although a wide range of non-motor symptoms (NMS) are frequently experienced by PD patients. One of the important and common NMS is cognitive impairment, which is measured using different cognitive scales. Monitoring cognitive impairment and its decline in PD is essential for patient care and management. In this study, our goal is to identify the most effective cognitive scale in predicting cognitive decline over a 5-year timeframe initializing clinical biomarkers and DAT SPECT.

### Methods

Machine Learning has previously shown superior performance in image and clinical data classification and detection. In this study, we propose to use machine learning with different types of data, such as DAT SPECT and clinical biomarkers, to predict PD-CD based on various cognitive scales. We collected 330 DAT SPECT images and their clinical data in baseline, years 2,3,4, and 5 from Parkinson's Progression Markers Initiative (PPMI). We then designed a 3D Autoencoder to extract deep radiomic features (DF) from DAT SPECT images, and we then concatenated it with 17 clinical features (CF) to predict cognitive decline based on Montreal Cognitive Assessment (MoCA) and The Movement Disorder Society-Unified Parkinson's Disease Rating Scale (MDS-UPDRS-I).

### Results

The utilization of MoCA as a cognitive decline scale yielded better performance in various years compared to MDS-UPDRS-I. In year 4, the application of the deep radiomic feature resulted in the highest achievement, with a cross-validation AUC of 89.28, utilizing the gradient boosting classifier. For the MDS-UPDRS-I scale, the highest achievement was obtained

**Data Availability Statement:** The Python script used for this research is accessible at the following URL: https://github.com/gorjiarman/PD-MCI-Prediction. The data included in the notebook were

procured directly from the Parkinson's Progression Markers Initiative (PPMI) and are subject to a Data Usage Agreement that prohibits public sharing or republishing. This dataset was collected by PPMI from 2010 through 2023. Our team, by Ali Fathi Jouzdani, accessed it in February 2021 and downloaded data up until February 2023 for this study. PPMI's dataset comprises anonymized imaging, clinical, and demographic information, ensuring that no personal identifiers, such as names or addresses, are disclosed. In compliance with the Data Usage Agreement, we are restricted from redistributing the data or publicly disclosing any patient identifiers. Access to the dataset is granted upon user registration at the PPMI website: https://www.ppmi-info.org/access-data-specimens/download-data. PPMI is listed in the National Institutes of Health's clinical trial registry under the identifier NCT04477785."

**Funding:** This work was funded by the Michael J. Fox Foundation for Parkinson's Research (MJFF-021134; https://www.michaeljfox.org). The funders had no role in study design, data collection and analysis, decision to publish, or preparation of the manuscript.

**Competing interests:** The authors have declared that no competing interests exist.

by utilizing the deep radiomic feature, resulting in a cross-validation AUC of 81.34 with the random forest classifier.

## Conclusions

The study findings indicate that the MoCA scale may be a more effective predictor of cognitive decline within 5 years compared to MDS-UPDRS-I. Furthermore, deep radiomic features had better performance compared to sole clinical biomarkers or clinical and deep radiomic combined. These results suggest that using the MoCA score and deep radiomic features extracted from DAT SPECT could be a promising approach for identifying individuals at risk for cognitive decline in four years. Future research is needed to validate these findings and explore their utility in clinical practice.

## Introduction

Parkinson's disease (PD) is a neurodegenerative disorder that affects the basal ganglia, an area of the brain that regulates movement [1]. However, PD also has a variety of non-motor symptoms (NMS) that are related to a mix of dopaminergic and non-dopaminergic pathways, showing PD's multi-focal and widespread pathology [2]. Cognitive problems are one of the most common and important NMS that can happen at any stage of the disease [3]. Cognitive impairment in PD can range from mild cognitive impairment to dementia, impacting various cognitive domains. The cognitive abilities of PD patients often decline throughout the disease, and it is increasingly recognized as a significant aspect of PD progression [4]. In the past few years, many studies have focused on predicting cognitive decline in PD (PD-CD) [5–10]. The prediction of PD-CD involves a multifaceted approach incorporating various markers, assessments, and biomarkers over different time frames [11–13]. These predictive models provide valuable insights for patient management, clinical trial design, and the development of treatments for PD-CD.

Different scales are utilized for the prediction of PD-CD, and various studies have delved into the predictive capabilities of these scales [14–16]. Montreal Cognitive Assessment (MoCA) [17] is one of the best-known cognitive scales widely used to measure cognitive aspects of PD [5,8,9]. MoCA was initially created to assess mild cognitive impairment linked to Alzheimer's Disease, focusing on areas such as memory, executive functions, and verbal fluency. Another scale is UPDRS which is a widely accepted tool for measuring the severity of PD and tracking changes in motor and non-motor function over time [18]. The Movement Disorder Society-sponsored revision of the UPDRS (MDS-UPDRS) is a clinical assessment tool introduced in 2008 as an updated version of the original UPDRS. The MDS-UPDRS is comprised of four sections, with Part I dedicated to assessing non-motor aspects of daily living such as assessment of cognitive impairment, hallucinations and psychosis, depressed mood, anxious mood, and apathy [19].

The presence of visual hallucinations and psychosis has been associated with an increased risk of cognitive decline and dementia in PD [20]. Also, mood disturbances are prevalent non-motor symptoms in PD and can significantly impact cognitive function [21]. Based on these results, we chose to utilize the overall score of MDS-UPDRS-I rather than solely focusing on the cognitive impairment question. (Further details on the MoCA and MDS-UPDRS-I, as well as their respective questionnaires, can be found in S1 Table).

Predicting PD-CD is feasible across various periods, as evidenced by prior research employing different durations [5–10]. For instance, one study demonstrated that PD-CD could be forecasted within 2 years by integrating factors such as age, non-motor evaluations, DAT imaging, and CSF biomarkers [22]. Another investigation revealed that close to half of the PD patients who initially presented with normal cognitive function developed PD-CD within 6 years [12]. Furthermore, another study has indicated that baseline pro-saccadic metrics are predictive of PD-CD over 4.5 years [23].

In addition, researchers are investigating the use of various biomarkers to anticipate the development of PD-CD. The link between cognitive problems and age and Postural Instability and Gait Disorders (PIGD) was previously discussed in another study [24,25]. Another study also found that age is a predictor of MDS-UPDRS-I [26]. Minor hallucinations in people with PD might be a predictor of faster PD-CD [27]. Prior research has established a link between olfactory impairment and cognitive impairment [28–31]. According to several studies, the gene variant apolipoprotein E (APOE) is associated with PD-CD [32,33]. Cerebrospinal Fluid (CSF) biomarkers can also be useful in the prediction of PD-CD, one study [34] demonstrated the possible involvement of tau species in the progression of cognitive symptoms in PD. Another study suggests that CSF α-synuclein levels correlate with PD-CD [35]. In addition to clinical data, several imaging techniques have been used to investigate changes in brain structure and function that may be predictive of future PD-CD. Dopamine active transporter (DAT), located in the presynaptic terminal of the dopaminergic projection and responsible for dopamine reuptake, is a marker of dopamine innervation, and a DAT SPECT is commonly used to diagnose dopaminergic neuron loss in the striatum [36]. Biomarker studies have revealed that lower dopamine levels on DAT SPECT, are linked to PD-CD [5,16,22,37]. Caspell-Garcia et al. found that lower DAT availability in the putamen is a longitudinal biomarker predictor of developing cognitive problems [38]. Another study reported that cognitive impairment was linked to lower DAT density on SPECT, especially in the caudate [39].

Traditionally, researchers have relied on classical statistical models to predict PD-CD. However, machine learning approaches are gaining traction due to their ability to handle complex datasets and potentially improve prediction accuracy [40–42]. Machine learning leverages clinical and imaging data more effectively, particularly in predicting PD-CD [6,43–45]. Unlike classical models, these approaches achieve high performance without the need for extensive manual programming [46].

PD-CD is a common and disabling symptom of PD, although its prediction and diagnosis are challenging due to the heterogeneity and complexity of the disease [47]. In this study, we used machine learning with different types of data, such as DAT SPECT and clinical biomarkers, to predict PD-CD based on two cognitive scales. We aim to compare the prognostic accuracy of MoCA and MDS-UPDRS-I based on the year and features that we use. Our approach will provide more accurate and personalized predictions of PD-CD and will contribute to the development of better diagnostic and therapeutic strategies.

## Materials and methods

In the below sections, we investigate our diverse data selection/generation, image processing methods, machine learning, and analysis methods. As shown in Fig 1, we first encoded our imaging data and clinical biomarkers, and then we used combinations of clinical and deep radiomic features to predict whether a PD-CD happens 2, 3, 4, and 5 years after the initial diagnosis using the MoCA and MDS-UPDRS-I scales. Unlike previous studies, here we defined PD-CD as any slight decrease in the score of MoCA or MDS-UPDRS-I, thereby enhancing our model's sensitivity to even subtle cognitive deteriorations.

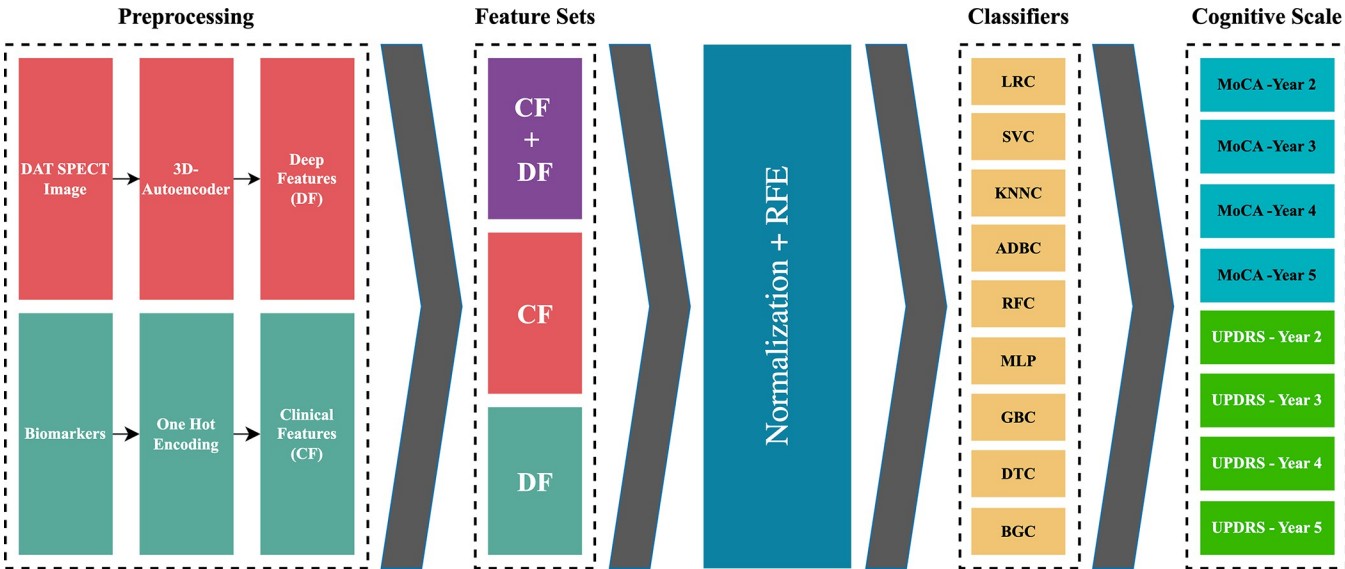

**Fig 1. Data collection procedure.** After extracting DFs, we combined CF and DF to make three different combinations of features. After feature set generation, we fed each feature set into nine different classifiers to predict PD-CD based on two cognitive scales and four timeframes. CF: Clinical feature, DF: Deep features, RFE: Recursive Feature Elimination, ADBC: AdaBoost Classifier, BGC: Bagging Classifier, SVC: Support Vector Classifier, KNN: K-Nearest Neighbors Classifier, RFC: Random Forest Classifier, GBC: Gradient Boosting Classifier, MLP: Multi-Layer Perceptron, DTC: Decision Tree Classifier, LRC: Logistic Regression Classifier, CF: Clinical Features, DF: Deep Features.

## Participants

Data for this study were retrieved from the Parkinson's Progression Markers Initiative (PPMI) database, adhering to previously established inclusion and exclusion criteria for PPMI [48]. We further filtered participants based on our inclusion criteria: 1) Participants were required to exhibit all 17 clinical features at baseline assessment. 2) Participants needed to complete MDS-UPDRS-I and MoCA scores at baseline and follow-up visits at years 2, 3, 4, and 5. 3) All participants included in the study underwent DAT SPECT imaging at baseline. Based on these criteria, we included 330 patients and their DAT SPECT and clinical data in baseline; 204 and 126 patients, respectively, were male and female; the average age at baseline was 61.2 ± 9.6 years.

## Clinical features

Different clinical variables and biomarkers are used to address cognitive impairment severity and prognosis. As previously discussed, recent studies showed that advanced age, genetic variation in APOE, gait disturbance, motor assessments, non-motor assessments, DAT imaging, electroencephalogram, and CSF biomarkers may contribute to the early prediction of PD-CI [5–7,10,24,37,38,44,49]. We used 17 of the most predictive clinical biomarkers in previous studies to further investigate the effect of clinical biomarkers alongside DAT SPECT. These biomarkers are age, CSF amyloid-β 42, CSF α-Synuclein, hallucination, MDS-UPDRS-III, PIDG, CSF P-tau, CSF T-tau, serum uric acid, disease duration, The Scale for Outcomes in Parkinson's disease for Autonomic symptoms (SCOPA-AUT), Geriatric Depression Scale (GDS), APOE genetic variation, gender, Orthostatic hypotension, diabetes, and hypertension [50–52]. We then used one hot encoding to encode all of these biomarkers to clinical features (CF).

## Deep radiomic features

To investigate DAT SPECT images, we conducted a deep radiomic feature extraction from the DAT SPECT images that were extracted from PPMI. We first segmented the dorsal striatum

on DAT SPECT before feature extraction, using the same procedures as in our earlier study [16] (S1 File and S1 Fig). Following the segmentation process, we cropped the DAT SPECT images according to the segmentations and finally resized each image to its final size of 32 X 32 X 32. Intensity normalization was a further step we used before feature extraction. We also augmented images just by flipping them from left to right, as suggested in previous studies [53]. Subsequently, the images were inputted into the autoencoder for feature extraction. A detailed explanation of our proposed 3D-Autoencoder can be found in the S1 File. Using this 3D-Autoencoder, we extracted 1024 DFs from the bottleneck layer using DAT SPECT images. To further explore the predictive performance of clinical and deep features combined, we created a third feature set from a combination of the extracted data called CF + DF.

## Machine learning models

Based on previous studies [5–10], we used nine classifiers, AdaBoost Classifier (ADBC) [54], Support Vector Classifier (SVC) [55], K-Nearest Neighbors Classifier (KNNC) [56], Random Forest Classifier (RFC) [57], Gradient Boosting Classifier (GBC) [58], Bagging Classifier (BGC) [59], Multi-Layer Perceptron (MLP) [60], Decision Tree Classifier (DTC) [61], and Logistic Regression Classifier (LRC) [62] (Fig 1). They were chosen experimentally from different families of learning algorithms. Several studies have shown that using only some of the most relevant features improves performance on a variety of tasks compared to using all of them. It has been shown that most classifiers often cannot cope with large amounts of input to work [63,64]. Therefore, it is important to choose the optimal subset of features to use as input to avoid overfitting. We used Recursive Feature Elimination (RFE) [65] to select the top 10, 50, and 100 features from each feature set. In addition, we tuned the hyperparameters of the classifier using 5-fold cross-validation and grid search optimization techniques. Grid search optimization is a powerful way to significantly improve the performance of your ML methods. The data points are divided into 4 convolutions for training and 1 convolution for testing in 5-fold cross-validation. Moreover, 80% of the training data points were used for cross-validation and 20% were used for external tests.

## Results

We selected 330 patients from PPMI that had baseline, year 2, year 3, year 4, year 5 MoCA, and MDS-UPDRS-I scores. Before deploying machine learning algorithms, we performed a statistical examination with an unpaired t-test to assess differences between PD-CD individuals and non-converting PD (PD-NC) participants. This analysis considered a range of cognitive scales and time frames, with the findings presented in S2 and S3 Tables. Subsequently, this dataset was utilized to train our machine learning models to predict potential declines in MoCA and MDS-UPDRS-I scores over time. In total, 648 distinct trajectories were tested, encompassing 2 scales, 4 years, 3 feature sets, 3 feature sizes, and 9 algorithms. The performance of several algorithms and datasets is displayed in Fig 2. The best accuracy was selected between different combinations of feature set sizes for each trajectory.

The utilization of MoCA as a cognitive decline scale yielded better performance in various years compared to MDS-UPDRS-I. As shown in Fig 2 In year 4, the application of the DF resulted in the highest achievement, with a cross-validation AUC of 89.2, Utilizing the GBC with 100 features. The external test of 89.8 confirmed our finding. In year 3, the best outcome was attained by employing LRC and DFs, with a cross-validation AUC of 78.06 using 50 features, although the external test was 52. Similarly, in year 5, the combination of LRC and DF produced a cross-validation AUC of 77.93 using 50 features and an external test of 55.7. Conversely, year 2 demonstrated the lowest performance, with the most favorable outcome

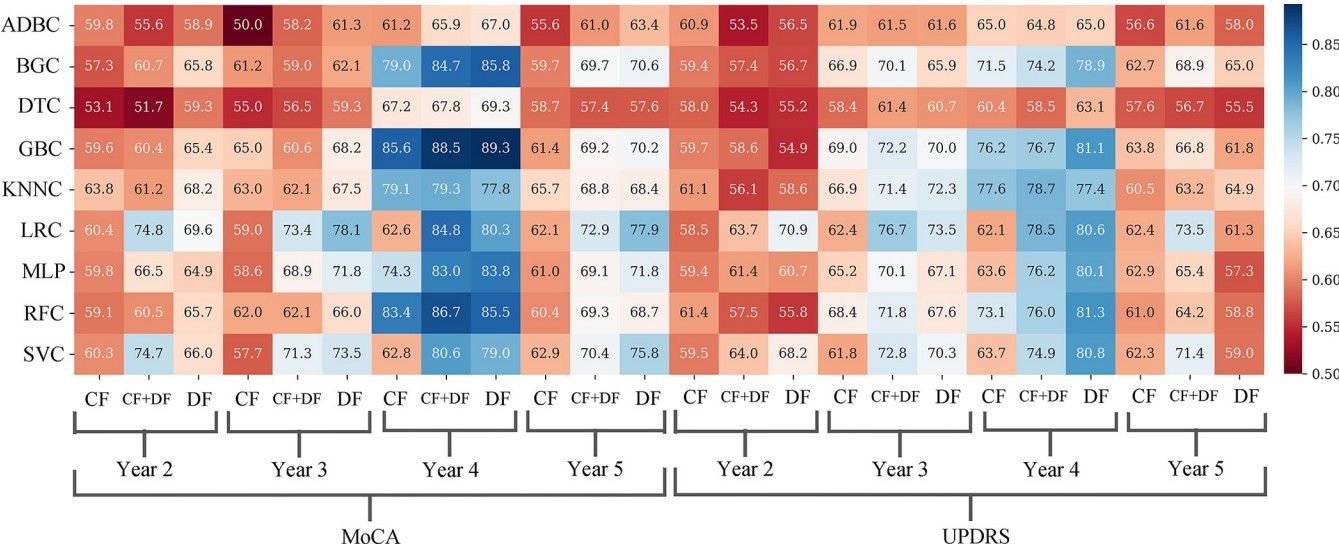

**Fig 2. Heatmap for cross-validation AUC scores.** Showing the superiority of MoCA performance compared to MDS-UPDRS-I and also the high performance of year 4 in both scales. ADBC: AdaBoost Classifier, BGC: Bagging Classifier, SVC: Support Vector Classifier, KNN: K-Nearest Neighbors Classifier, RFC: Random Forest Classifier, GBC: Gradient Boosting Classifier, MLP: Multi-Layer Perceptron, DTC: Decision Tree Classifier, LRC: Logistic Regression Classifier, CF: Clinical Features, DF: Deep Features.

achieved by utilizing both CF + DF, in conjunction with LRC, resulting in a cross-validation AUC of 74.79 using 50 features and an external test of 46.6.

The optimal performance when using MDS-UPDRS-I as the cognitive decline scale varied across different years, and similar to MoCA, As shown in Fig 2, year 4 had the best performance with the highest achievement obtained by utilizing the DF, resulting in a cross-validation AUC of 81.34 with the RFC using 100 features and an external test of 78.6 confirmed our finding. For year 3, the best result was achieved through the use of LRC and DF, yielding a cross-validation AUC of 76.73 and an external test of 61.9 using 50 features. Similarly, in year 5, the combination of LR and DF + CF led to the best outcome, with a cross-validation AUC of 73.52 and an external test of 51.3. On the other hand, year 2 exhibited the lowest performance, with the most favorable result achieved by employing DF and LRC, resulting in a cross-validation AUC of 70.90 using 50 features with an external test of 43.5. External test results are shown in S3 Fig.

The MoCA and MDS-UPDRS-I scale ROC curves for the fourth year are displayed below in Fig 3. MoCA outperforms MDS-UPDRS-I in PD-CD prediction for all three feature sets.

## Discussion

The primary aim of this study was to evaluate the accuracy of the MOCA and MDS-UPDRS-I in predicting cognitive decline in PD over five years, utilizing ML techniques. Moreover, the study aimed to investigate the potential of different features, such as DAT SPECT imaging and clinical variables, to improve the accuracy and reliability of predicting PD-CD.

Cognitive impairment in PD can significantly affect the quality of life, functional independence, and treatment outcomes for both patients and their caregivers [66]. It is uncertain how cognitive decline in PD will ultimately manifest. Patients may either fully regain their cognitive abilities, remain in a state of mild cognitive impairment, or develop dementia [67]. Also, PD-CD is influenced by a variety of factors, including motor symptoms, non-motor symptoms, medical comorbidities, and psychosocial conditions, all of which interact in complex

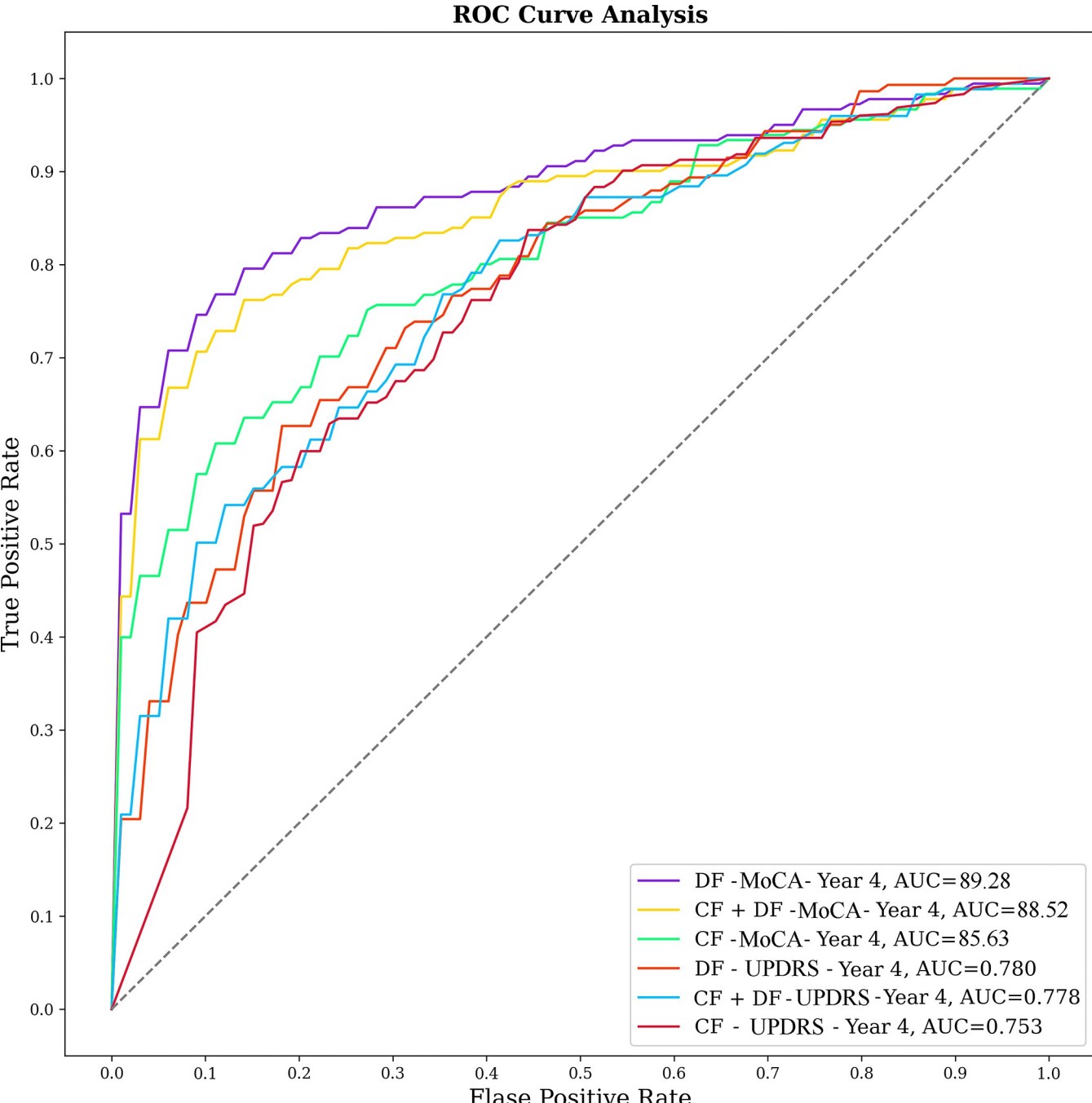

**Fig 3. Illustrates the ROC analysis.** Showcasing the superior results for both the MoCA and MDS-UPDRS-I across all three feature sets. The peak AUC performance for the MoCA was secured through the deployment of DF for forecasting the MoCA score in year 4. Similarly, the optimal AUC for the MDS-UPDRS-I was also obtained by employing DF to estimate the MDS-UPDRS-I score in the same year. CF: Clinical feature, DF: Deep features.

ways [68]. The research conducted by Weil et al. emphasizes the importance of understanding mild cognitive impairment in PD and its progression to dementia [69]. This knowledge can pave the way for early interventions targeting cognitive decline in PD patients. Moreover, Ray et al. suggest that identifying patients at risk of cognitive decline early in the course of PD can aid in stratifying individuals for targeted interventions [70].

Cognitive assessment scales are essential for evaluating cognitive function in PD patients and detecting mild cognitive impairment and dementia, as well as cognitive decline for monitoring stages, which are common in PD. In our most recent research [16], we utilized HMLS for forecasting cognitive impairment in the fourth year, relying exclusively on the MoCA score as the cognitive measure without reference to alternative scales. In contrast, our current objective is to evaluate the MoCA against other cognitive scales present in PPMI, specifically MDS-UPDRS-I, and we found that using changes in MoCA score as a metric for cognitive decline can be more predictable compared to MDS-UPDRS-I. To our knowledge, this comparison was not done elsewhere before. One of the key distinctions between the MoCA and the MDS-UPDRS-I lies in their assessment focus. Although both measure NMS, the MoCA questionnaire evaluates a broader range of cognitive functions, while the MDS-UPDRS-I assesses symptoms related to cognition (S1 Table). Also, The Montreal Cognitive Assessment (MoCA) test is designed to be administered by a healthcare professional [17], But the MDS-UPDRS-I questionnaire is structured so that it can be filled out independently by the patient, collaboratively with caregiver input, or solely by the caregiver, based on the preference of the patient and caregiver [19].

Notably, the most significant results were observed in year four, suggesting that the onset of PD-CD becomes apparent and can be predicted with high accuracy at this stage. Previous studies indicated that PD-CD can become noticeable within 5 years of diagnosis [71]. This period is characterized by a decline in various cognitive domains, including executive function, attention, memory, and visuospatial abilities. Identifying these changes early on can aid in implementing appropriate interventions and support strategies to manage cognitive decline effectively.

DAT SPECT, which plays a significant role in tracking dopamine in the brain, could serve as a valuable tool for monitoring PD-CD. As shown in our study, the deep features extracted from DAT SPECT imaging offer a promising avenue for providing a comprehensive dataset that surpasses traditional clinical data in insight extraction. This sophisticated imaging technique serves as a quantitative biomarker for assessing the onset and progression of PD. Furthermore, the use of radiomic analysis on longitudinal DAT SPECT images has been shown to improve the prediction of PD outcomes, underscoring the diagnostic value of DAT SPECT imaging in this context [72].

One of the challenges we encountered in our study was the lack of longitudinal data from various cognitive questionnaires for temporal analysis. This shortfall in suitable longitudinal data necessitated the exclusion of two questionnaires namely Site Investigators Decision (SID) and MDS Task Force Guideline (MDS-TFG) neurophysiological battery. We also were not able to study the MDS-UPDRS-I cognitive impairment question individually as the PD-CD and PD-NC groups are extremely (1:5) imbalanced and even with implementing the Synthetic Minority Over-sampling Technique (SMOTE) our results were significantly overfitted.

As mentioned, we utilized DAT SPECT scan images due to their relevance to the dopamine-based mechanisms of PD. However, other imaging modalities may yield better results, and further studies are needed to compare the efficacy of different modalities. Looking ahead, exploring additional biomarkers and conducting longitudinal analyses of other medical recordings through machine learning techniques could greatly advance the development of AI models that are closely aligned with clinical applications; for example, the use of mobile and tablet-based applications for the ongoing longitudinal monitoring of patient's cognitive functions, coupled with the integration of these data into artificial intelligence tools, holds substantial promise for the field [73]. This allows us to have access to more data and train more robust models.

## Conclusions

The study findings indicate that the decline in MoCA as a measure for PD-CD results in higher performance within 5 years compared to MDS-UPDRS-I especially in year 4. Furthermore, deep radiomic features had better performance compared to sole clinical biomarkers or clinical and deep radiomic combined. These results suggest that using the MoCA score and Deep Radiomic features extracted from DAT SPECT could be a promising approach for identifying individuals at risk for cognitive decline in four years. Future research and additional data are needed to validate these findings and explore their utility in clinical practice.

## Supporting information

**S1 File. Imaging protocols.** Consist of three parts 1) S1 Appandix DAT SPECT image characteristics 2) S1 Protocol Image Preprocessing 3) S2 Protocol 3D-Autoencoder.
(DOCX)

**S1 Fig. Preprocessing steps.** 1) Smooth images 2) Increase the contrast of the images 3) Use a threshold to digitize the image 4) Crop 3D ROI.
(TIFF)

**S2 Fig. 3D autoencoder architecture.** It has four convolutional layers, each followed by a batch normalization and max-pooling operation. The pooling layers are used to reduce the number of parameters. The decoder path has four convolutional layers, each followed by batch normalization.
(TIFF)

**S3 Fig. Heatmap for external test AUC based on different trajectories explained in the method section.** ADBC: AdaBoost Classifier, BGC: Bagging Classifier, SVC: Support Vector Classifier, KNN: K-Nearest Neighbors Classifier, RFC: Random Forest Classifier, GBC: Gradient Boosting Classifier, MLP: Multi-Layer Perceptron, DTC: Decision Tree Classifier, LRC: Logistic Regression Classifier, CF: Clinical Features, DF: Deep Features.
(TIFF)

**S1 Table. Detail of MDS-UPDRS-I and MoCA questionnaires.** In the MDS-UPDRS-I, we utilized a summation of five sections that assess cognitive-related symptoms in Parkinson's Disease (PD). The total response to these items is scored on a scale of 0 to 20. Additionally, itemw from the MoCA is referenced in the accompanying table. The total response to these items is scored on a scale of 0 to 30. MDS-UPDRDS-I: The Movement Disorder Society-Unified Parkinson's Disease Rating Scale, MoCA: Montreal Cognitive Assessment.
(DOCX)

**S2 Table. Paired T-test outcomes for the MDS-UPDRS-I score.** In year 2, there was a significant gender-related discrepancy observed between the PD-CD and PD-NC groups. Year 3 highlighted a substantial difference in PIGD scores between the same groups. However, in years 4 and 5, the analysis revealed no significant differences between PD-CD and PD-NC.
(DOCX)

**S3 Table. Results of the paired T-test for the MoCA score.** In year 2, a notable difference was observed in T-tau levels between the PD-CD and PD-NC groups. Year 3 saw a significant disparity in gender distribution between the two groups. No substantial differences were detected in year 4. However, in year 5, significant variations were found in a-synuclein, P-tau, T-tau, and diabetes occurrences between the PD-CD and PD-NC groups.
(DOCX)

## Author Contributions

**Conceptualization:** Arman Gorji, Ali Fathi Jouzdani.

**Investigation:** Arman Gorji, Ali Fathi Jouzdani.

**Methodology:** Ali Fathi Jouzdani.

**Software:** Arman Gorji, Ali Fathi Jouzdani.

**Supervision:** Ali Fathi Jouzdani.

**Visualization:** Arman Gorji, Ali Fathi Jouzdani.

**Writing – original draft:** Arman Gorji, Ali Fathi Jouzdani.

**Writing – review & editing:** Ali Fathi Jouzdani.

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
