## [Decision Letter · Decision Letter 0]

7 Feb 2024

PONE-D-23-28754Comparing Cognitive Assessment Scales for Predicting Cognitive Decline in Parkinson's Disease: A Hybrid Machine Learning Approach Using DAT SPECT and Clinical BiomarkersPLOS ONE

Dear Dr. Fathi Jouzdani,

Thank you for submitting your manuscript to PLOS ONE. After careful consideration, we feel that it has merit but does not fully meet PLOS ONE’s publication criteria as it currently stands. Therefore, we invite you to submit a revised version of the manuscript that addresses the points raised during the review process.

**ACADEMIC EDITOR: **Thank you for submitting your manuscript to PLOS ONE. After careful consideration by 2 Reviewers and an Academic Editor, all of the critiques of both Reviewers must be addressed in detail in a revision to determine publication status. If you are prepared to undertake the work required, I would be pleased to reconsider my decision, but revision of the original submission without directly addressing the critiques of the Reviewers does not guarantee acceptance for publication in PLOS ONE. If the authors do not feel that the queries can be addressed, please consider submitting to another publication medium. A revised submission will be sent out for re-review. The authors are urged to have the manuscript given a hard copyedit for syntax and grammar.

We look forward to receiving your revised manuscript.

Kind regards,

Stephen D. Ginsberg, Ph.D.

Section Editor

PLOS ONE

Journal Requirements:

“Parkinson’s Progression Markers Initiative (a public-private partnership) is funded by the Michael J Fox Foundation for Parkinson’s Research and funding partners, including AbbVie, Allergan, Avid Radiopharmaceuticals, Biogen, BioLegend, Bristol-Myers Squibb, Celgene, Denali, GE Healthcare, Genentech, GlaxoSmithKline, Lilly, Lundbeck, Merck, Meso Scale Discovery, Pfizer, Piramal, Prevail Therapeutics, Roche, Sanofi Genzyme, Servier, Takeda, Teva, UCB, Verily, Voyager Therapeutics, and Golub Capital. Data used in the preparation of this article were obtained from the Parkinson’s Progression Markers Initiative (PPMI) database (www.ppmi-info.org/data).**”**

“No-  The funders had no role in study design, data collection and analysis, decision to publish, or preparation of the manuscript.”

Reviewers' comments:

Reviewer's Responses to Questions

**Comments to the Author**

1. Is the manuscript technically sound, and do the data support the conclusions?

Reviewer #1: Partly

Reviewer #2: Partly

2. Has the statistical analysis been performed appropriately and rigorously? 

Reviewer #1: No

Reviewer #2: Yes

3. Have the authors made all data underlying the findings in their manuscript fully available?

Reviewer #1: Yes

Reviewer #2: Yes

4. Is the manuscript presented in an intelligible fashion and written in standard English?

Reviewer #1: No

Reviewer #2: Yes

5. Review Comments to the Author

Reviewer #1: In the manuscript entitled “Comparing Cognitive Assessment Scales for Predicting Cognitive Decline in Parkinson's Disease: A Hybrid Machine Learning Approach Using DAT SPECT and Clinical Biomarkers”, the authors incorporated a special technique to visualize the dopamine levels in the brain region, especially striatum along with the machine learning. However, the manuscript does not seem to be novel as there are lots of publications that are available on PUBMED related to the same approach. For instance, Mahdi Hosseinzadeh et al., 2023, Hannes Almgren et al., 2023, Hojoong M Kim et al., 2019, Mohammad Salmanpour et al., 2020, and many others. Additionally, the manuscript hold the space for improvement as it contains several flaws, which should be rectified before any decision. The writing and arrangement of the manuscript is also not up to the mark, and thus, the authors need to improve it.

1. The authors in the manuscript mentioned that they involved the PD patients in a 4-year interval. This is hard to understand why they use 4-year intervals and what is the rationale behind this selection. They should use other time intervals. It is noteworthy to mention that if they use different time intervals then it will be a good comparative study.

2. In the introduction section, more focus should be on HMLSs and its relationship with DAT SPECT. Here, the authors need to elaborate the previous studies that involved theses two techniques and how theses techniques contribute to PD diagnosis.

3. In the section clinical data, the authors mentioned that they involved only 5 clinical biomarkers. The most important question is why they neglect other important biomarkers, such as gait abnormalities, which is one of the most important characteristic features of PD. The authors should mention the inclusion and exclusion criteria in the entire manuscript in a different subsection to increase the readability of the manuscript.

4. Further, the authors incorporated APOE genetic variation. However, studies demonstrated that genetic variation in α-Synuclein is related to PD more extensively. Thus, the authors should use genetic variation in α-Synuclein against the genetic variation in APOE.

5. The discussion and conclusion section seems to be weak. It should be elaborated more. Further, the authors should mention the drawbacks and challenges of the current manuscript along with the implementation of ML in PD diagnosis.

Reviewer #2: The Manuscript (PONE-D-23-28754) entitled Comparing Cognitive Assessment Scales for Predicting Cognitive Decline in Parkinson's Disease: A Hybrid Machine Learning Approach Using DAT SPECT and Clinical Biomarkers” aims to evaluate the combination of cognitive assessment scales and biomarkers which may provide an accurate prediction of cognitive decline in Parkinson's disease (PD) patients.

Authors indicated that the combination of clinical biomarkers and imaging data improved the accuracy of cognitive decline prediction in PD.

In general, the Manuscript is not well written and clear to understand, consequently it requires some major revisions.

In addition, objectives and the rationale are not clearly stated in the Manuscript.

The organization of the Manuscript is confusing, not very rational, and inappropriate in some sections.

Specific comments:

Cognitive impairment is a common non-motor symptom in PD which is frequently associated with olfactory deficit as indicated in previous studies (Solla et al., 2023, https://doi.org/10.3390/biology12010112; Baba et al., 2012; Fang et al., 2021; Cecchini et al., 2019, https://doi.org/10.1007/s00702-019-01996-z). Authors should include this finding in the Introduction section.

The Introduction section is too long and should be summary and focused on the specific topic. It is not necessary to describe the MoCA test, the Movement Disorder Society-Unified Parkinson's Disease Rating Scale (MDS-UPRDS-I) and other scales in the Introduction. These sections should be moved in the Methods section.

In the Participant section Authors should indicate the exact number of men and women enrolled.

The main limitations of this Manuscript should be included at the end of the Discussion section.

The conclusions need to be implemented with future perspectives.

6. PLOS authors have the option to publish the peer review history of their article (what does this mean?). If published, this will include your full peer review and any attached files.

Reviewer #1: **Yes: **Dr. Rohan Gupta

Reviewer #2: No

---

## [Author Response · Author response to Decision Letter 0]

28 Apr 2024

Response to Reviewers

Reviewer #1: 

In the manuscript entitled “Comparing Cognitive Assessment Scales for Predicting Cognitive Decline in Parkinson's Disease: A Hybrid Machine Learning Approach Using DAT SPECT and Clinical Biomarkers”, the authors incorporated a special technique to visualize the dopamine levels in the brain region, especially striatum along with the machine learning. However, the manuscript does not seem to be novel as there are lots of publications that are available on PUBMED related to the same approach. For instance, Mahdi Hosseinzadeh et al., 2023, Hannes Almgren et al., 2023, Hojoong M Kim et al., 2019, Mohammad Salmanpour et al., 2020, and many others. Additionally, the manuscript hold the space for improvement as it contains several flaws, which should be rectified before any decision. The writing and arrangement of the manuscript is also not up to the mark, and thus, the authors need to improve it.

- Thank you so much for your consideration. In 2023, Mahdi Hosseinzadeh and colleagues utilized the MoCA cognitive scale in the fourth year to forecast cognitive deterioration. In a similar timeframe, Hannes Almgren and his team also employed MoCA for anticipating cognitive shifts over four years. Earlier, in 2019, Hojoong M. Kim and associates conducted a comparative analysis of MoCA, DRS-2, and MMSE as cognitive assessments over an average span of 3.8 years. Subsequently, in 2020, Mohammad Salmanpour and his group undertook a clustering task that centered on the motor outcomes of patients, applying the HMLS method. Contrasting with these studies, our study predicts cognitive decline based on MoCA and the MDS-UPDRS-I score. Moreover, to enhance the precision of predicting Parkinson's Disease-Cognitive Decline (PD-CD), our study incorporated assessments over various durations.

1. The authors in the manuscript mentioned that they involved the PD patients in a 4-year interval. This is hard to understand why they use 4-year intervals and what is the rationale behind this selection. They should use other time intervals. It is noteworthy to mention that if they use different time intervals then it will be a good comparative study.

- We appreciate your comment. As previously stated in our last paper, the four-year mark was selected based on prior findings indicating it as the pivotal year when cognitive abilities typically diminish, allowing for more accurate predictions. However, your idea about the concept of comparing various time intervals was fascinating, as it promised a more in-depth analysis of performance for each cognitive scale. As a result, in our revised study, we assessed different time frames, specifically 2, 3, 4, and 5 years, which adds another novelty to our work. We decided against including additional years because they would significantly reduce our data sample size, hindering the effective application of machine learning algorithms.

2. In the introduction section, more focus should be on HMLSs and its relationship with DAT SPECT. Here, the authors need to elaborate the previous studies that involved theses two techniques and how theses techniques contribute to PD diagnosis.

- Thank you so much for your insightful comments. To distinguish our research from earlier studies, we shifted the emphasis of our paper from HMLS to an analysis of cognitive scales and timeframes. We employed machine learning techniques in place of HMLS to simplify the analysis of trajectories. Additionally, we referenced prior research that utilized ML or HMLS for forecasting cognitive outcomes based on DAT SPECT or clinical data in the introduction section of our paper.

3. In the section clinical data, the authors mentioned that they involved only 5 clinical biomarkers. The most important question is why they neglect other important biomarkers, such as gait abnormalities, which is one of the most important characteristic features of PD. The authors should mention the inclusion and exclusion criteria in the entire manuscript in a different subsection to increase the readability of the manuscript.

- In our study, we initially selected five clinical biomarkers to enhance performance, as suggested by the referenced research (Predictors of cognitive impairment in Parkinson’s disease: a systematic review and meta‑analysis of prospective cohort studies). Based on your comment and upon further examination of additional literature and a deeper consideration of your feedback, we opted to incorporate additional clinical biomarkers, including gait abnormalities and fifteen other characteristics. Furthermore, we have detailed inclusion and exclusion criteria for participants in the participants section of our paper.

4. Further, the authors incorporated APOE genetic variation. However, studies demonstrated that genetic variation in α-Synuclein is related to PD more extensively. Thus, the authors should use genetic variation in α-Synuclein against the genetic variation in APOE.

- We appreciate your comments. While cerebrospinal fluid α-Synuclein is more extensively connected to the risk of Parkinson's disease development, APOE variations have been shown to be more predictive of cognitive decline, as indicated by research (Alpha-Synuclein and Cognitive Decline in Parkinson's Disease). In light of your feedback and our study's evolution towards a multi-omics methodology and cognitive-related clinical biomarkers, we have included α-Synuclein in our clinical features to enhance the predictive accuracy of our models.

5. The discussion and conclusion section seems to be weak. It should be elaborated more. Further, the authors should mention the drawbacks and challenges of the current manuscript along with the implementation of ML in PD diagnosis.

- Based on your valuable comments, we have significantly overhauled our manuscript. The discussion section has been thoroughly revised to provide an in-depth analysis of various cognitive scales and time intervals. Additionally, we have comprehensively outlined the principal challenges and limitations encountered in our research. Furthermore, with the shift in emphasis away from HMLS, we have succinctly referenced prior studies that have employed ML to predict PD-CD in the introduction section.

Reviewer #2: 

The Manuscript (PONE-D-23-28754) entitled Comparing Cognitive Assessment Scales for Predicting Cognitive Decline in Parkinson's Disease: A Hybrid Machine Learning Approach Using DAT SPECT and Clinical Biomarkers” aims to evaluate the combination of cognitive assessment scales and biomarkers which may provide an accurate prediction of cognitive decline in Parkinson's disease (PD) patients.

Authors indicated that the combination of clinical biomarkers and imaging data improved the accuracy of cognitive decline prediction in PD.

In general, the Manuscript is not well written and clear to understand, consequently it requires some major revisions.

In addition, objectives and the rationale are not clearly stated in the Manuscript.

The organization of the Manuscript is confusing, not very rational, and inappropriate in some sections.

Specific comments: Cognitive impairment is a common non-motor symptom in PD which is frequently associated with olfactory deficit as indicated in previous studies (Solla et al., 2023, https://doi.org/10.3390/biology12010112; Baba et al., 2012; Fang et al., 2021; Cecchini et al., 2019, https://doi.org/10.1007/s00702-019-01996-z). Authors should include this finding in the Introduction section.

- Thank you for your comment. We have included the mentioned references and used their findings. 

The Introduction section is too long and should be summary and focused on the specific topic. It is not necessary to describe the MoCA test, the Movement Disorder Society-Unified Parkinson's Disease Rating Scale (MDS-UPDRS-I) and other scales in the Introduction. These sections should be moved in the Methods section.

- We greatly appreciate your insightful comments. Following your suggestions, we have refined the introduction to better reflect our unique approach, which emphasizes the comparative analysis of cognitive scales and temporal intervals. To enhance the readability of the introduction, we have moved the comprehensive details of the scales into the supplementary materials.

In the Participant section Authors should indicate the exact number of men and women enrolled.

- We added the exact number of men and women. 

The main limitations of this Manuscript should be included at the end of the Discussion section.

- We have included a discussion of the limitations in the relevant section.

The conclusions need to be implemented with future perspectives.

- We have included a discussion of the future perspectives in the relevant section.

---

## [Decision Letter · Decision Letter 1]

9 May 2024

Machine Learning for Predicting Cognitive Decline within Five Years in Parkinson's Disease: Comparing Cognitive Assessment Scales with DAT SPECT and Clinical Biomarkers

PONE-D-23-28754R1

Dear Dr. Fathi Jouzdani,

We’re pleased to inform you that your manuscript has been judged scientifically suitable for publication and will be formally accepted for publication once it meets all outstanding technical requirements.

Kind regards,

Stephen D. Ginsberg, Ph.D.

Section Editor

PLOS ONE

**Comments to the Author**

1. If the authors have adequately addressed your comments raised in a previous round of review and you feel that this manuscript is now acceptable for publication, you may indicate that here to bypass the “Comments to the Author” section, enter your conflict of interest statement in the “Confidential to Editor” section, and submit your "Accept" recommendation.

Reviewer #1: All comments have been addressed

2. Is the manuscript technically sound, and do the data support the conclusions?

Reviewer #1: Yes

3. Has the statistical analysis been performed appropriately and rigorously? 

Reviewer #1: Yes

4. Have the authors made all data underlying the findings in their manuscript fully available?

Reviewer #1: Yes

5. Is the manuscript presented in an intelligible fashion and written in standard English?

Reviewer #1: No

6. Review Comments to the Author

Reviewer #1: Accepted in its current form. The manuscript significantly contribute to the scientific faterenity and may open a door for future studies.

7. PLOS authors have the option to publish the peer review history of their article (what does this mean?). If published, this will include your full peer review and any attached files.

Reviewer #1: **Yes: **Rohan Gupta

---

## [Editor Report · Acceptance letter]

12 Jun 2024

PONE-D-23-28754R1 

PLOS ONE

Dear Dr. Fathi Jouzdani, 

I'm pleased to inform you that your manuscript has been deemed suitable for publication in PLOS ONE. Congratulations! Your manuscript is now being handed over to our production team.

Kind regards, 

on behalf of

Dr. Stephen D. Ginsberg 

Section Editor

PLOS ONE